# An Assessment Approaches and Learning Outcomes in Technical and Vocational Education: A Systematic Review Using PRISMA

**Siti Raudhah M. Yusop [1], Mohammad Sattar Rasul [1,*], Ruhizan Mohamad Yasin [1], Haida Umiera Hashim [2]** and **Nur Atiqah Jalaludin [1]**

[1] Faculty of Education, Universiti Kebangsaan Malaysia, Bangi 43600, Malaysia; p109257@siswa.ukm.edu.my (S.R.M.Y.); ruhizan@ukm.edu.my (R.M.Y.); nuratiqah.jalaludin@ukm.edu.my (N.A.J.)

[2] English and Linguistics Department, Academy of Language Studies, Universiti Teknologi Mara, Shah Alam 40450, Malaysia; haidaumiera@gmail.com

*   Correspondence: drsattar@ukm.edu.my

**Abstract:** Technical and vocational education and training (TVET) assessments give precise feedback on whether students have successfully attained learning outcomes. It can improve teaching quality and empower students, educators, and stakeholders to take action. Only a few studies have attempted to review the literature on this topic systematically. The PRISMA (preferred reporting items for systematic reviews and meta-analysis) criteria are used to guide this systematic review, which uses the following three key databases: Web of Science (WoS), Scopus, and Google Scholar. A thorough search of the electronic database utilising keywords and search strings yielded 78 publications published between 2015 and 2021, with 29 studies related to the topic highlighted. The Mixed Methods Appraisal Tool is used to evaluate all of the chosen articles (MMAT). The findings reveal that students' learning outcomes are frequently examined utilising a competency-based assessment technique in TVET. Competence is used to assess students' learning results, and it is advised that student competency development be prioritised.

**Keywords:** assessment; learning outcomes; competency; technical and vocational

## 1. Introduction

Technical and Vocational Education and Training (TVET) plays an essential role in ensuring the country's economic viability by facing the challenges of globalisation and acting in line with the development of the K-economy. Furthermore, the adoption of the UNESCO Education Strategy for the period 2014–2021, as well as the Sustainable Development Goals (SDGs) adoption and the Education 2030 agenda, occurred in a context of globalisation and rapid technological development, which were characterised by changes in economic, labour market, and skill patterns [1,2]. They also reported that labour of advanced skills, quantity and quality of skills, access to education, and occupational profile are some of the critical current trends and practises in TVET mainly. The TVET curriculum structure emphasises practical orientation, developing soft skills needed for decent jobs, and preparing students for the labour market [3]. TVET students are given the training to acquire the knowledge, skills, and attitudes set during their studies. Determining learning outcomes, effective delivery of knowledge, skills, competencies, and assessment of student achievement are essential aspects of the empowerment of TVET in shaping a dynamic and effective education system. Competence refers to a person's performance to perform a task effectively, especially when necessary to play a role or achieve a task, mission, or be measurable [4]. Assessment of student competence in written or practical form is essential to confirm that students have demonstrated mastery of practical knowledge and skills and

the essential ability required to perform tasks according to specific curriculum standards in TVET [5].

Thus, competence is knowledge, skills, attitudes, and a dynamic concept for taking the following actions [6]. Students need the achievement of good competencies to further their studies and shape their future, meeting the needs of the world of work. For a country, the achievement of good competencies is required to produce a workforce that can contribute to technical and vocational fields. However, assessment in measuring student achievement, especially in skills, is still lacking, especially in TVET [7]. According to the interviews conducted, the assessment approach utilised was restricted to traditional assessment, and the concept is likewise new to the majority of educators [8]. This issue has an impact on the learning outcomes that students are expected to achieve in terms of knowledge, skills, and competencies [9]. Therefore, the purpose of this review paper is to provide an overview of TVET assessment. The researchers intended to delve deeper into the assessment approaches used by educators and the learning outcomes expected from the assessment, particularly in terms of developing student competencies. We are also interested in the assessment techniques that are often implemented in TVET, which are significant given the current challenges in research on education. Therefore, this systematic study was conducted to answer the following questions:

1. What are the assessment approaches used to assess students in technical and vocational education?
2. What are the intended student learning outcomes of the educators' assessment?

*The Needs for a Systematic Literature Review Related to Student Learning Outcome from Assessment Approaches in TVET*

Assessment is part of the learning process [10]. It describes the reputation of the school, school district, and school personnel based on the test scores [11]. There is also a growing concern about properly evaluating employees in the workforce, such as through performance appraisals [12]. Student assessment of learning outcomes is to see how well they match what the educators intended. After the process, some judgement is usually made when grades or marks are placed [13]. Although related articles abound, very few have attempted to review these articles systematically, limiting future scholars' ability to identify changes and new opportunities in the emerging literature [14,15]. Furthermore, there is a requirement to provide current knowledge in order to ensure that significant insights can be derived from the existing literature. This is because, even though the importance of assessment in student learning has become increasingly recognised over the last three decades, it remains to impact externally conducted accountability and high-importance certification examinations, indicating a need for quality assessment in educators' assessment practise [16–18]. Furthermore, curricular modifications in vocational education and training emphasise curriculum design with objectives or efficiency [19]. Assessments must specify what they are to be used for and how they will be used. They must also employ appropriate procedures for efficiently gathering and interpreting information, determining competency through assessment, and recording and reporting evaluation outcomes to stakeholders [20]. In conjunction with this, the current study aims to conduct a systematic literature review on students' learning outcomes in TVET assessment.

In the context of this study, a systematic review can be defined as an examination of a formulated question that employs systematic and explicit methods to identify, select, and critically appraise relevant research studies, as well as analyse their data [21]. Systematic reviews also seek to investigate secondary data by obtaining, synthesising, and evaluating existing information on a subject in a logical, clear, and analytical manner [22]. Conducting a systematic review of this issue is vital since there is a new issue of discourse concerning it. The method implemented in this review allows for identifying gaps and determining the direction for future research related to how educators approach assessment and respond to student learning outcomes. Based on the analysis in SLR and meta-analysis, it can show the trends, identify gaps, and results from comparison [14,23]. A systematic review also allows

researchers to identify patterns in previous studies and helps them understand related issues that can provide insights into various assessments administered by educators. It is expected to be able to clearly identify assessment learning outcomes such as students' knowledge, skills, competencies, and attitudes.

## 2. Materials and Methods

This systematic literature review was conducted using the preferred reporting items for systematic review and meta-analysis (PRISMA) guideline method. According to [14], the preparation of articles using the PRISMA method includes the following three phases of preparation: identification, screening, and inclusion. The researchers began by developing research questions and locating articles on assessment and the construction of student learning outcomes, competencies, and performances. The researchers also discussed the appraisal of quality, data extraction, and analysis.

### 2.1. Preferred Reporting Items

One of the most critical areas of education research is a systematic review of the assessment approach used by TVET teachers and the learning outcomes that can be generated. This systematic review is guided by the PRISMA publication standard, which identifies research entry criteria and large amounts of scientific database literature [24].

### 2.2. Formulation of Research Questions

As the primary goal of this article was to systematically review the existing literature related to assessment approaches in the context of educators and the learning outcome intended, the study must develop a suitable research question that guides the entire systematic review methodology during the initial phase of the review. The development of student knowledge, skills, competencies, and attitudes is an example of a learning outcome. The following research question was chosen: what assessment approaches are used to evaluate students in technical and vocational education, and what are the intended learning outcomes? Following the formulation of the research question, the primary focus of this article was on the assessment approaches used by educators in the TVET sector.

### 2.3. Systematic Searching Strategy

The researchers' search strategy included the following three subprocesses: identification, screening, and eligibility.

#### 2.3.1. Identification

Identification is a procedure that is used to improve the significance of the keywords that are employed. This is significant since the identifying procedure enhances the likelihood of receiving more relevant articles for the review [24]. This systematic literature review used the following three familiar database identities: Web of Science (WoS), SCOPUS, and Google Scholar. WoS and SCOPUS are two competitive and always up-to-date world-class databases. Google Scholar is a database that allows for independent and extensive research in a variety of fields. It is also easy to access. According to [25], Google Scholar has greatly extended its coverage over the years, making it a robust database of scholarly literature. WoS was launched in 1997 and rebranded as the Web of Science Core Collection around 2014. It was initially managed by the Institute for Scientific Information but is currently managed by Clarivate Analytics. On the other hand, SCOPUS is the highest-level peer-assessed journal in the academic, government, and corporate fields around the world related to the fields of medicine, science, and humanities. Over the past 15 years, more than 3000 journals have been published by WoS. More than 2500 journals have been published by SCOPUS with the fields of medicine, general, and internal, are the highest publications of the two databases. WoS, on the other hand, offers broad participation in a wide scope of academic subjects.

This comparative study also found that most of the WoS and SCOPUS articles are related to bibliometrics studies and meta-analysis [26]. Both of the databases are widely used as the domain searches related to health sciences, medicine, information science, and libraries, as well as science. Google Scholar was used by the researchers to search for any related articles using the exact keywords used in Scopus and WoS, as well as, whenever appropriate, searching techniques such as Boolean operators (AND, OR, NOT or AND NOT), phrase searching, truncation and wildcard ("*") and field code functions (either combining these search techniques or using them separately) relying on the search efforts. Furthermore, the researchers conducted manual searches, selecting relevant articles from SCOPUS, WoS, and Google Scholar. Table 1 shows the keywords used when looking for articles related to the assessment in technical and vocational. Additional information such as literature type, language, and the timeline were entered according to the researchers' criteria is shown in Table 2.

**Table 1.** Keywords used for the process of finding relevant literature.

| Databases | Keywords Used | Identification Phase | Included Phase |
|---|---|---|---|
| SCOPUS | TITLE-ABS-KEY (("assessment" OR "competency * assessment" OR "performance * assessment") AND ("student * learning outcome *" OR "student * performances *" OR "students * competency *" AND ("technical and vocational education" OR "technical education" OR "vocational education" OR "TVET")) | 35 | 14 |
| WoS | TS = (("assessment" OR "competency * assessment" OR "performance * assessment") AND ("student * learning outcome *" OR "student * performances *" OR "students * competency *" AND ("technical and vocational education" OR "technical education" OR "vocational education" OR "TVET")) | 28 | 10 |
| Google Scholar | Using specific keywords from Scopus and WoS, as well as Boolean operators, phrase searches, and field code functions (either together or individually) as appropriate | 15 | 5 |
| | Publications earned | 78 | 29 |

**Table 2.** The eligibility and exclusion criteria.

| Criterion | Eligibility | Exclusion |
|---|---|---|
| Literature type | Journal (research articles) | Book, book series, chapter in book, systematic review articles, conference proceeding |
| Language | English | Non-English |
| Timeline | Between 2015 and 2021 | 2014 and earlier |
| Country/territory | World | |

### 2.3.2. Screening Phase

At this phase, researchers carefully identified duplicate articles in SCOPUS, WoS, and Google Scholar. A total 78 articles were gathered from SCOPUS, Web of Science (WoS), and Google Scholar database, respectively. In total, four duplicate articles were identified, and 42 articles were finalised to be brought to the next phase of this systematic literature review. The remaining 42 articles were examined in detail to meet the criteria set by researchers. The selected articles must pass the criteria set by researchers, such as the type of literature, language, and period. The selection of SCOPUS, WoS, and Google Scholar indexed journals is as described in the identification phase in Section 2.1. Data are obtained from a nationwide study, and the writing is in English to avoid translation problems. The study used a seven-year time span (i.e., publications published between 2015 and 2021) since it offered a sufficient number of articles for review. Furthermore, we established the most recent time span in order to acquire the most recent data and research on TVET assessment. Further discussion is summarised in Table 2.

### 2.3.3. Eligibility

The second screening stage was used to check that all of the articles that survived the first screening phase fit the standards. Throughout this stage, the articles were re-evaluated for their suitability for the review based on the title and abstract. If the writers were still perplexed by the contents, they decided to explore the contents of the selected articles. At this phase, only articles that passed all requirements in the two phases and met the eligibility criteria were selected by researchers. The aspects of exceptions include books, book series, chapters in books, systematic review articles, conference proceedings, non-English language articles, and those published after 2015 are included to obtain accurate and quality data. Because the review focused on mixed research designs (quantitative + qualitative + mixed methods), the quality of the selected publications was assessed using the Mixed Methods Appraisal Tool (MMAT) version 2018 [27]. The final articles are about 29 articles included in these systematic reviews. The work steps are shown in tabular form in Figure 1.

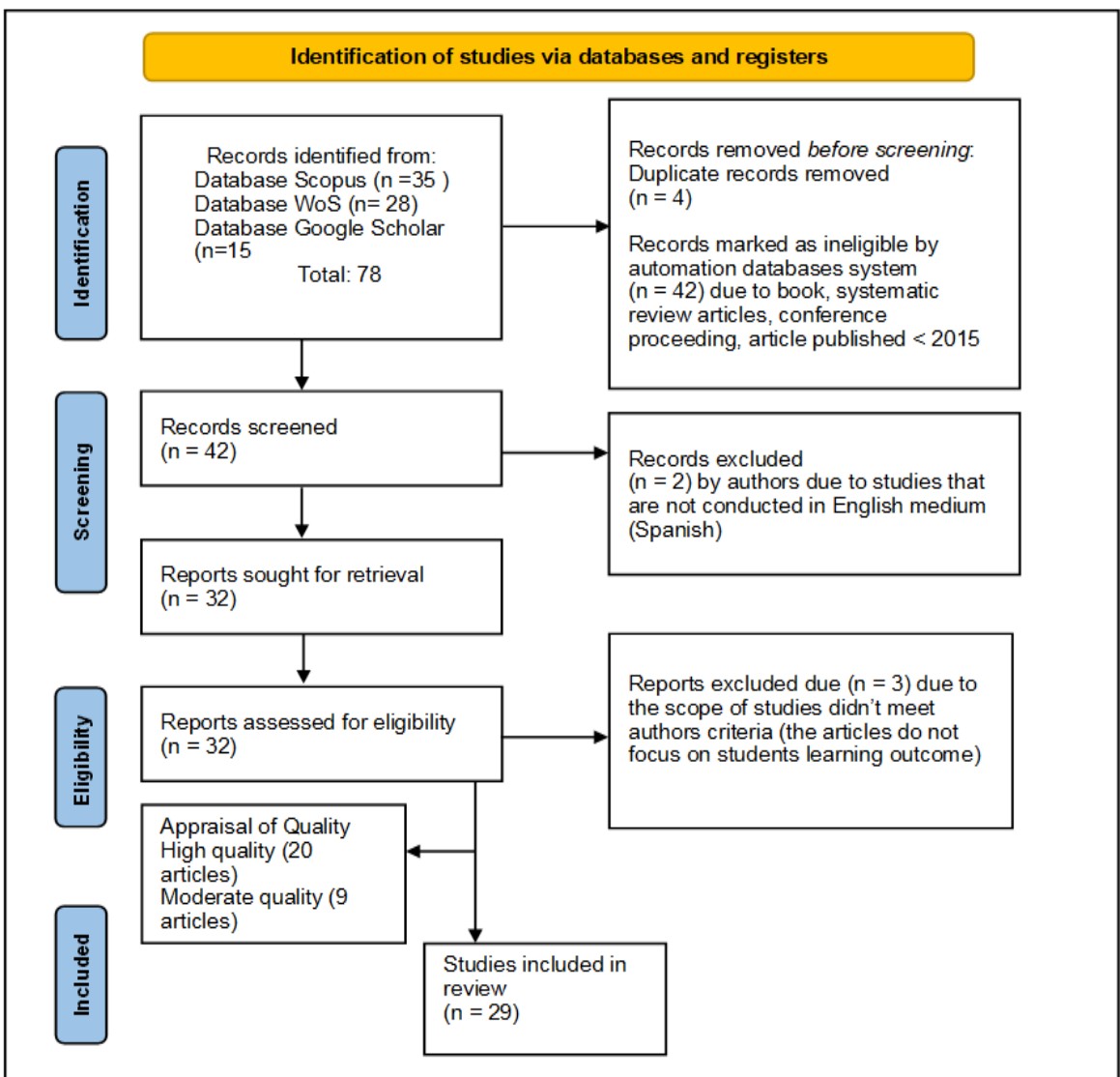

**Figure 1.** The stream chart of the examination adapted from [28]. Click or tap here to enter text.

### 2.4. Quality Appraisal

The quality of the selected publications was assessed using the Mixed Methods Appraisal Tool (MMAT) version 2018. This systematic review assigned three reviewers to

evaluate the quality of the selected articles based on the clarity of the research questions, confidence in the assessment of the research questions, sampling, data collection methods, and suitability of the statistical analysis performed to achieve the objective. Furthermore, two reviewers examined how the data in the articles were interpreted and the presentation of results, discussion, and conclusion. The MMAT guidelines were used to determine the quality, with 25% accounting for low-quality articles, 50% average, 75% above average, and 100% high; analysis is the qualitative analysis of mixed. The reviewers then classified twenty (20) articles as having high average quality, while the remaining nine (9) were above average.

### 2.5. Data Extraction and Analysis

The remaining papers were evaluated and analysed. The researchers were required to extract all relevant data from the selected papers before analysis, with the study's research question guiding the procedure. Because the review focused on the primary and empirical data of the selected prior research, we first looked for the relevant three areas of these papers, namely, the abstract, findings, and discussion, before moving on to the other sections to search for any related material. The retrieved data were organised into a table to facilitate the synthesis process. A qualitative synthesis was carried out using thematic analysis to uncover themes connected to assessment in students' learning outcomes and TVET assessment. Thematic analysis is a process for identifying, analysing, organising, describing, and reporting themes discovered in extracted data [29]. For academics, theme analysis has several advantages. First, it evaluates several views, discovering similarities and contrasts while also providing new discoveries. The strategy aids in summarising the primary topic of large amounts of data, encouraging researchers to use a well-structured way to manage the data, resulting in clear and organised findings [21]. More crucially, theme analysis is more suited to the nature of our current evaluation, which focuses on mixed research designs [30]. Thematic analysis is a synthesis approach for research designs that involve both qualitative and quantitative investigations, according to [24,31]. Patterns have to be found based on the retrieved data to generate themes. The researchers created themes by identifying any parallels or correlations between the retrieved data. Changes in TVET provide a greater focus on components of curricular content, such as learning outcomes or competency building [32]. The situation helps construct the themes needed in this study.

The procedure was carried out by two coders, who created the following four key themes: TVET assessment approaches, students' competencies development during an assessment, students' performance development, and positive impact of assessment. The researchers then conducted a thematic analysis to identify acceptable subthemes in each generated theme, yielding two themes. The coders double-checked the correctness of the main themes and subthemes to confirm their accuracy and relevance to the study's research goals. Any dispute amongst the coders on the created themes or subthemes was discussed, and external expert opinions were sought if the coders could not achieve a mutual consensus. The experts would next decide on the subtheme most suited for this study. Several subthemes were renamed and re-located inside the four significant themes throughout this process.

Finally, we identified two primary themes (students' assessment approaches and learning outcomes) and 15 subthemes, which are competency-based assessment, performance-based assessment, formative assessment, criteria-based assessment, e-portfolio assessment, school-based assessment, summative assessment, workplace assessment, computer-based assessment, inclusive-based assessment, classroom-based assessment, scenario-based assessment for assessment technique, and competency development, performance development and positive impact of assessment for students' learning outcomes themes. Coding for the theme and subtheme shown in Tables 3 and 4.

**Table 3.** The findings (the themes and subthemes) elaboration.

| Authors/Country | Main Study Design | Assessment Approaches | Students Learning Outcome | | |
| --- | --- | --- | --- | --- | --- |
| | | | Competency Development | Performance Development | Positive Impact of Assessment |
| Ibrahim Mokhtar et al., (2017)—Malaysia | QL | FA | | | 1. Enhance learning (EL) 2. Succeed in achieving learning outcomes (SLO) 3. Build positive attitude (PA) |
| Okolie et al., (2020)—Nigeria | QL | CBA | Employability skill (ES) | | 1. Integrating theory and practice (ITP) 2. Enhance learning (EL) 3. Allow students to construct learning on their own (CL) |
| Hegarty et al., (2019)—New Zealand | MM | E-PA | | | 1. Build confident (BC) 2. Build networking (BN) 3. Reduced student fail (RSF) |
| Ana et al., (2019)—Indonesia | MM | CBA | Technical competency (TC) Generic competency (GC) Industrial competency (IC) | | |
| Mohamed et al., (2021)—Malaysia | QN | CBA | Work skill competency (WSC) | | |
| Revilla-Cuesta et al., (2020)—Spain | MM | FA | Professional competencies (PC) | | |
| Abdul Musid et al., (2019)—Malaysia | QL | CBA | Technical skill (TC) soft skill (GC) | | |
| Slogar et al., (2021)—Croatia | QN | CBA | Leadership skills (LS) entrepreneurial competencies (ECS) | | |
| Levanova et al., (2020)—Rusia | QL | CRBA | Project competency (PC) | | |
| Yamada et al., (2018)—Ethiopia | QN | PBA | | Comprehensive skills (CS) | |
| Hashim et al., (2019)—Malaysia | MM | SBA | | | Develop knowledge construction process (KCP) |
| Nzembe (2018)—South Africa | QL | SA | | | 1. Students were happy in the way they were assessed (AT) 2. They expressed satisfaction on the way they are assessed (AT) |
| Kim et al., (2019)—Rwanda | QN | CBA | Developing content knowledge skill (CKS) and Instructional skills (IS) | | |
| Mazin et al., (2020)—Malaysia | MM | CBA | Learning competency (LC) | | Learning analytics (LA) |
| Hui et al., (2017)—Malaysia | QN | PBA | | Progressive performance and technology innovation performances (PTIP) | |
| Mohd Ali et al., (2019)—Malaysia | QN | PBA | | Students' performance in Mathematic (SPM) | Reduces students' anxiety towards math (AT) |
| Ab Rahman et al., (2020)—Malaysia | MM | PBA | | Student performance in item laboratory assessment (PLA) | |
| Bekri et al., (2015)—Malaysia | QN | CBA | | | Development of 4 domains; in ICT Skills (ICTS) |
| Dogara et al., (2020)—Nigeria | QN | FA | Soft skills and teamwork skills (GS) | | |
| Sugiyanto et al., (2020)—Indonesia | MM | CBA | Project competency including (PC) | | |
| Dahlback et al., (2020)—Norway | QL | WBA | Teaching practice competency (TPC) | | Integrating both theory and practice (ITP) |
| Gulikers et al., (2018)—The Netherlands | QL | CBA | | | 1. Student learned a lot from the assessment (EL) 2. Student motivational for their preparation for VET (AT) |
| Rausch et al., (2016)—German | QN | COMA | Problem solving competency (PSC) | | |
| Seifried et al., (2020)—German | QN | PBA | Problem solving competency (PSC) | | |
| Lee at al. (2020)—Malaysia | QL | CBA | Teaching practice competency (TPC) | | |
| Ewing et al., (2017)—Australia | MM | SCBA | Competency in mathematics and numeracy (SPM) | | |
| Nkalane (2018)—South Africa | QL | IA | | | Student's success in learning (EL) |
| Avdarsol et al., (2020)—Kazakhstan | QL | CRBA | | | Student's functional literacy in computer science (ICTC) |
| Sephokgole et al., (2019)—South Africa | QN | CLBA | | | Student knowledge and skills to meet real life situation (BKS) |

QN = quantitative; QL = qualitative; MM = mix method; CBA = competency-based assessment; PBA = performance-based assessment; FA = formative assessment; CRBA = criteria-based assessment; E-PA = e-portfolio assessment; SBA = school-based assessment; SA = summative assessment; WBA = workplace assessment; COMA = computer-based assessment; IA = inclusive-based assessment; CLBA = classroom-based assessment; SCBA = scenario-based assessment.

**Table 4.** An assessment approaches and student's learning outcome. The themes and subthemes—coding.

| CBA | | PBA | | FA | CRBA | E-PA | SBA | SA | WBA | COMA | SCBA | IA | CLBA |
|---|---|---|---|---|---|---|---|---|---|---|---|---|---|
| [33–42] | | [43–47] | | [48–50] | [51] | [52] | [53] | [54] | [55] | [56] | [57] | [58] | [59] |
| ES ITP EL CL TC GC LS ECS CKS IS LC | LA ICTS GS PC EL AT TPC IC WSC PC TC GC | CS PTIP SPM | AT PLA PSC | EL AT | PC ICTS | BC BN RSF | KCP | ATA | TPC ITP | PSC | SPM | EL | BKS |

### 2.6. Strength and Limitation

Our review utilised the PRISMA guidelines to find as many suitable studies as possible. We widened our search keywords and databases and actively discussed any inconsistencies. Despite our intention to provide an international component to our analysis, we elected to limit our search to two databases known for their quality and contribution to research to ensure the rigour and quality of the papers included in our assessment. We emphasised the quality of the articles chosen above the scope of the study; unfortunately, this resulted in the selection of just 29 publications from 14 nations. Given this result, one wonders if, by using additional databases, more research from a more extensive range of countries could not have been included.

### 3. Results

After going through four stages of identifying the eligible articles for this systematic literature review's analysis, 29 publications pertaining to the assessment and creation of TVET student learning outcomes were discovered. According to this review, competency-based assessment is the most often employed assessment technique for those evaluating students' learning outcomes in TVET. Of the 29 articles that have been reviewed, the analysis obtained has found that, from the assessment approach, the intended student learning outcomes areas are in Table 3.

### 3.1. The Selected Studies' General Context

Regarding the selected studies, 10 studies focused on assessment and student learning outcomes in Malaysia, 3 studies in South Africa, 2 studies in Nigeria, Germany, Indonesia, and 1 study in Croatia, Ethiopia, Rwanda, New Zealand, Australia, Norway, the Netherlands, Spain, Russia and Kazakhstan. Moreover, there were 11 quantitative research studies, 10 qualitative research studies, and 8 mixed-method (qualitative and quantitative) studies selected for the review.

### 3.2. The Assessment Approaches and Students' Learning Outcomes

This section focuses on the assessment approaches practices by educators in TVET sector, based on the 2 primary themes (students' assessment approaches and learning outcomes) and 15 subthemes; competency-based assessment, performance-based assessment, formative assessment, criteria-based assessment, e-portfolio assessment, school-based assessment, summative assessment, workplace assessment, computer-based assessment, inclusive-based assessment, classroom-based assessment, scenario-based assessment, competency development, performance development, and positive impact of assessment.

*3.3. Assessment Approaches Used to Assess Students in Technical and Vocational Education*

Assessment is usually conducted for various purposes depending on the stakeholders to obtain information that can improve students' achievement and improve teachers' teaching. The change in an assessment provides a clear picture of the world of research to determine the assessment that dominates nowadays. The relevant assessment approach in TVET exists in various terminologies, such as competency-based assessment and performance appraisal. This terminology conceptually has the same meaning, yet there are differences between each other. In TVET, performance appraisal is used to measure students' competence. Performance refers to the methods used by teachers to deliver teaching, knowledge, and skills, while competence emphasises minimum standards by adding levels of criteria, value orientation, and quality of movement [43]. Therefore, performance is based on the focus on set objectives and competence is based on criteria.

This section presents the study's findings on the following first research question: What are the assessment approaches used to assess students in TVET? A competency-based assessment is a way to measure competency for a vocational skill [60]. Twelve studies have clearly described the competency-based assessment approach as the basis of their study, i.e., the study conducted by [33–42,61,62]. The assessment approach through performance-based assessment is also often used in assessing student learning outcomes. In this literature review, there are five studies related to performance-based assessment, namely [43–47]. In the formative assessment approach, there are three studies, namely [48–50]. The rest are studies using a variety of assessment approach terms, namely, criteria-based assessment [51], e-portfolio assessment [52], school-based assessment [53], summative assessment [54], workplace assessment [55], computer-based assessment [56], scenario-based assessment [57], inclusive-based assessment [58], and classroom-based assessment [59]. This summary can be seen as a whole in Figure 2.

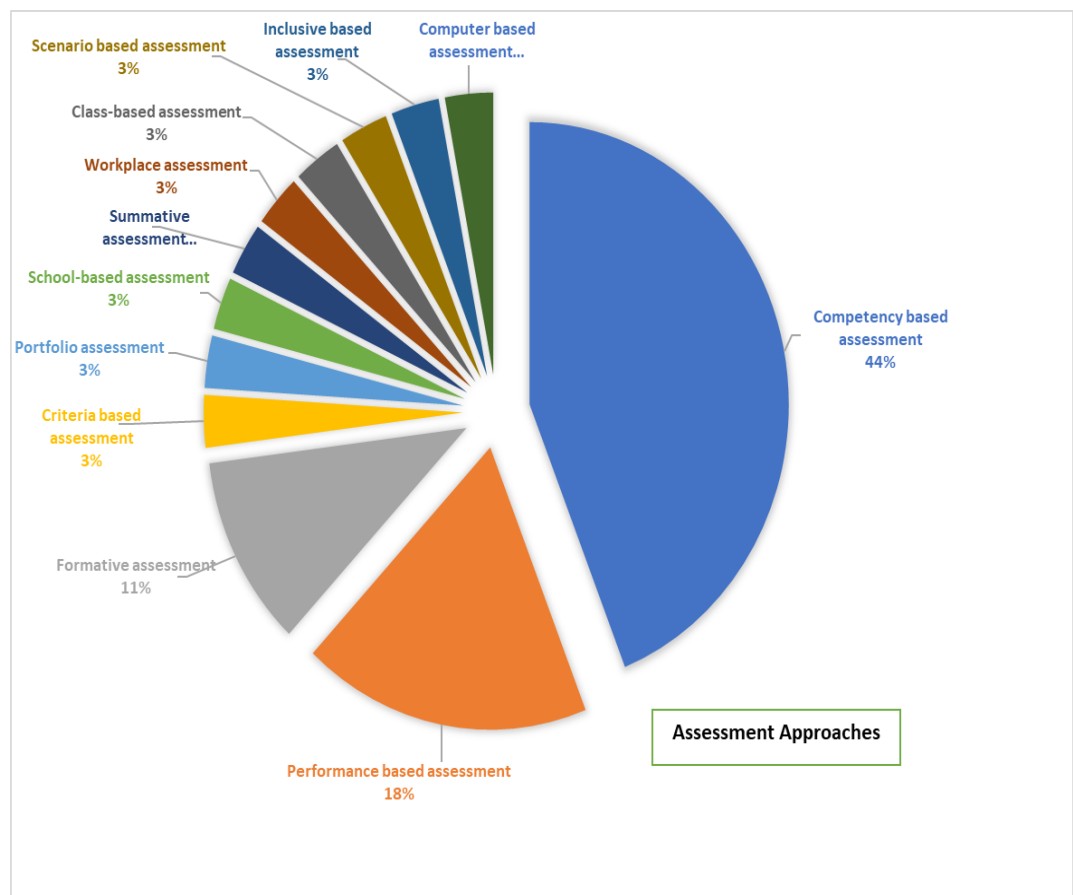

**Figure 2.** Assessment approaches in TVET.

### 3.3.1. Competency-Based Assessment (CBA)

Studies on competency-based assessment have been linked to learning methods. According to the findings, CBA is associated with problem-based learning [60]. Problem-based learning (PBL) integrates theory and practice among students, motivates students, and enables students to build their learning during assessment [61,62]. PBL has also succeeded in improving the competencies and abilities of students. However, the main challenges in implementing PBL and competency assessment are a lack of facilities for teacher teaching and learning, unqualified teachers, and corruption. Competency-based assessments conducted through work-based learning [38] show that work-based learning has a strong relationship with the development of teamwork skills among students. Work-integrated learning (WIL) has a significant negative relationship with the development of teamwork skills among students. WIL also showed a non-significant direct positive relationship with an assessment technique, according to a CBA study [38]. Assessment techniques also had a significant indirect positive relationship with the development of teamwork skills for students in Indonesia's primary schools [39]. Studies on mobile learning and project-based learning in CBA show that combining mobile learning models with project-based learning can significantly improve students' cognitive, psychomotor, and affective domain competencies. Through full teacher monitoring, evaluation, and guidance, students perceive positive teacher involvement.

The study conducted by [33] in looking at differences in students' workplace skills based on gender, economic status, and academic and vocational achievement showed no significant differences in the variables studied. Students' perceptions of teacher competence are also moderate, according to the study. It also shows that students' workplace skills are at a moderate level. Teachers recommend being active in integrating workplace skills efficiently. A study conducted by [35] examined students' entrepreneurial competencies from the components of decision making, initiative, achievement, leadership, empathy, collaboration, integrity, and their relationship with students' socio-demographics. A comparative study of the implementation of CBA in the assessment of internship programmes from the aspects of standards, planning, methods, evidence, and judgement in three universities by the authors of [62] who showed that technical competencies are most often used by the three universities, namely, UPI (Indonesia), UNY (Indonesia), and NUoL (Laos).

Generic skills competencies were also implemented by all three universities. However, for industry competencies, only UPI and UNY use it. The project-based internship assessment method showed UPI using 84%, UNY 54%, and NUoL 32%. Overall, universities in Indonesia are successful in evaluating internship programmes in the industry, which is 78% compared to Laos, which is 35.5%. Some problems were identified, such as a lack of clear guidelines, a lack of mechanisms to coordinate quality assurance between institutions, and limited access to work with the industry. Studies on the aspect of student learning styles in CBA can be seen in the study of online courses. Students in massive open online courses are more likely to use knowledge assessment than practical assessment. However, the skills assessment shows a decrease of 2.7% compared to the knowledge assessment [37].

Students are more motivated to continue their studies in vocational fields. The study was conducted by [40] at the pre-vocational secondary level and found that students relate to this assessment in a more meaningful and realistic way than previous types of assessment. Instructors play a crucial role in implementing assessment successfully. However, the level of teaching competence of TVET trainees as a whole is still unknown [36]. There is a proposal to expand the programme to a larger scale so that it can help improve the quality of teachers' instructional skills and employees' teaching skills professionally. The competence of trainee teachers in teaching is determined by affective, psychomotor, and cognitive aspects [41]. Overall, the mean score of teacher trainee competence from affective aspects is 4.33 out of 5.0 (mean M = 3.99).

The study by the authors of [34] in identifying the constructs and elements of the assessment rubric for on-the-job training (OJT) succeeded in obtaining 110 elements suitable for training evaluation rubrics in the industry based on the course objectives intended for

students. An e-portfolio looks at the factors and indicators of e-learning based on expert agreement. The categories are the following: achievement recognition, virtual learning spaces, competency assessments, and operational systems. Ref. [42] have found that four categories have been identified for assessing the effectiveness of online education in Malaysia. The elements are achievement recognition, virtual learning spaces, competency assessments, and operating systems.

### 3.3.2. Performance-Based Assessment

Workers are highly skilled in sewing, ironing, and product finishing activities, but not very skilled in analysing garment structures or manufacturing patterns. Ref. [46] developed a unique instrument to track employee knowledge and skills in a natural work environment. Findings revealed significant positive relationships between pattern and structure (46.27%) and sewing and finishing (40.83%). The study conducted by [47] in applying graph theory to map the domain of student learning outcomes further developed the Malaysian Qualification Framework (MQF). This study involves correlation analysis and a regression test to measure the frequency of performance as a significant positive effect on the development of the framework.

Past studies show that assessment can be done with the help of technology. One of them is with the use of a computer. Computer-assisted assessment of problem-solving skills is a specialised domain in TVET. At the commercial level, it focuses on the understanding and competencies characteristic of the current field of work. The study of [44] aimed to develop computer-based assessments for low-level domains. The authors of [45] conducted a study on computer-adaptive assessment (CAT), and their results have shown successful impacts when compared to traditional assessment. CAT can reduce examiners' anxiety towards mathematics examinations compared to the traditional test approach at polytechnics in northern Malaysia that offer diploma certification in engineering. CAT can reduce students' anxiety about math tests and help improve student performance. This study shows that CAT for algebraic assessment is proven to be feasible and an easy-to-use tool. The study conducted by [43] on the other hand, analysed the effect of evaluator severity on student performance in installing electronic circuit project components using many-facet Rasch measurement (MFRM). This study showed that students have high competence in answering the rubric provided compared to the severity of the raters and complex items.

### 3.3.3. Formative Assessment

The study by [50] collected information on assessment practises that use a mastery learning approach. The teacher gives feedback, corrections, and continuous styling activities to students aimed at bridging learning gaps between current student knowledge and what they should know next. Vocational teachers use Mastery of Learning with a mobility module system. Students are fully responsible for completing assignments and detecting their mistakes after the teacher's feedback. This approach also encourages the attitude of helping each other to learn among students.

Revilla-Cuesta, Skaf, Manso and Ortega-López, [49] in their study aimed to assess students' perceptions of formative and cooperative assessments for the first time involving technical subjects. This assessment is related to several things, namely, the following: t The usefulness detected by the students in this teaching modality. The optional teaching modalities for this type of subject. Formative assessment is when students can correct each other and is relevant for optimal learning. Almost all subjects can be adapted to incorporate formative assessment and cooperative work teaching modalities. They are indirectly succeeding in promoting basic skills for the professional field. Teacher and student autonomy in carrying out an assessment is a crucial element to solving the practical aspects satisfactorily. The main form of assessment is formative assessment.

Project competence is an indicator of personal development as a result of mastery of project activities. It is expressed by possessing knowledge, skills, and exceptional knowledge

abilities. The authors of [48] found that assessment methods included operational feedback, continuous quality assessment for each completed practical, and criteria-based assessment.

### 3.3.4. Criteria-Based Assessment

The study of [51] aims to investigate the role of criteria-based assessment in developing students' functional literacy in computer science. The effectiveness of developing students' functional literacy is demonstrated through practical experiments and the proposed criteria-based assessment.

### 3.3.5. E-Portfolio Assessment

A vocational lecturer's facilitation influenced the engagement of fifteen carpentry students during their learning [52]. Students were asked to create a visual assessment e-Portfolio using smartphones and mobile applications. The majority of students found the apps (Evernote, Facebook, and G+) easy to access and use, and they felt safe doing so. Students believed that mobile learning aided their learning and provided them with opportunities to connect with others.

### 3.3.6. School-Based assessment

The process of constructing knowledge in an open learning system can help TVET practitioners complete a task, according to [53]. The TVET practitioner produced 211, or 92.95 per cent of the message code, with the theme of giving an opinion accounting for the highest frequency. The second-highest message code was 51 or 22.47 per cent.

### 3.3.7. Summative Assessment

The study of [54] investigated academic factors that either impede or promote access, participation, and success in a South African TVET college. A majority agreed that the college uses class tests, assignments, examinations, and practical projects such as Integrated Summative Assessment Tasks (ISAT). Academic factors, such as students' lack of preparedness for the TVET curriculum and the language of instruction, have the potential to make or break students' chances of accessing, participating in, and succeeding in their academic programmes.

### 3.3.8. Workplace Assessment

Authentic assessment of teachers leads to greater coherence between theory and practice. It also aids in the job-related learning process that leads to comprehensive teacher competence. This type of assessment, i.e., an authentic exam, necessitates an understanding of teachers' complex roles within their VET professional context. The authors of [55] investigated whether a workplace-based comprehensive exam in vocational teacher education at the university/PET level would offer students the opportunity to demonstrate their teaching competency.

### 3.3.9. Computer-Based Assessment

The authors of [56] conducted a study that assessed higher-order outcomes of vocational education and training (VET). They created a computer-based assessment of domain-specific problem-solving competence. The multidimensional Rasch analysis revealed satisfactory EAP/PV reliabilities ranging from 0.78 to 0.84 for the 'knowledge application' facets and 0.77 to 0.85 for the non-cognitive facets. Furthermore, the achievement differences between industrial clerks and their comparison groups are as expected in terms of their modelled problem-solving skills.

### 3.3.10. Scenario-Based Assessment

The cognitive diagnostic assessment task (CDAT) revealed students' difficulties in specific math areas and some difficulties with reading, comprehension, and understanding their questions. The authors of [57] project aims to investigate indigenous students' numer-

acy learning to gain practical knowledge about instructional and assessment approaches that can be used to improve student employment. The findings highlighted three different assessment approaches used to conclude students' competencies in VET courses. The study contributes to the literature on a holistic approach to assessing competency by providing a more comprehensive view of students' capabilities.

### 3.3.11. Inclusive-Based Assessment

The assessment practises designed by lecturers do not reflect student diversity, according to a study by [58] in South Africa. Students enrol in colleges to gain vocational skills rather than develop employability skills. The study examined the measures of inclusive assessment practices that are thought to create opportunities for all students of TVET colleges.

### 3.3.12. Classroom-Based Assessment

Classroom tests do not assess the knowledge and skills needed in the workplace or in real-life situations, according to [59]. The impact of WIL is negative because graduate students are not prepared to enter the labour market. To accommodate the wide range of students, a different type of assessment should be used.

### 3.4. Development of Learning Outcomes of TVET Assessment

Assessment is part of the education system which is aimed at measuring students' ability in setting the domain of learning outcomes that have been set. TVET assessments are conducted in the form of written and practical assessments aimed at assessing students' knowledge based on cognitive domains, students' skills based on psychomotor domains, and students' attitudes based on affective domains [37]. Therefore, this part presents the study's findings based on the second research question, namely, what are the intended student learning outcomes of the educators' assessment?

### 3.4.1. Building Student Competencies

Learning outcomes in the form of competency building is a positive approach desired in TVET. Competencies can help develop students' careers and then place students in a healthy competitive environment in the world of work if the acquired competencies meet the needs of the industry and current conditions. The study of [61] is related to employability skills construction, [62] is associated with the structure of technical competence, generic competence and industry competence, [33] is related to students' workplace competence, [34] is related to technical skills and soft skills, [35] is related to leadership skills and entrepreneurial competencies, [39,48] are related to project competencies that include the domain of knowledge, skills, and abilities, [37] is related to learning competence, [38] is related to soft skills construction and teamwork skills, [41,55] are related to teaching practice competence for teacher students covering the cognitive, psychomotor and affective domains; [44,56] are related to problem solving competence, and [57] pertaining to student competence in mathematics and numeracy.

### 3.4.2. Formation of Student Performances

Learning outcomes in the form of performance achievement are also an essential aspect in building the careers of TVET students. A study of student performance formation by [47] is related to progressive performance and technology innovation performance; Ref. [45] is related to students' performance in mathematics; Ref. [43] is related to student performance in item laboratory assessment; Ref. [46] is related to comprehensive skills.

### 3.4.3. Formation of the Impact of Assessment

Students' learning outcomes generally show that assessment plays a role in shaping the positive effects obtained by students. Typically, assessments performed by teachers create a positive learning environment, increase motivation and skills, improve learning outcomes

among students, and develop the process of building knowledge and skills needed in real life [40,50,53,58,59,61]. Assessment is also seen to be able to form a positive attitude in students, such as being responsible, building confidence, helping each other, having a happy and satisfying perspective on the assessment conducted, and forming a good personality [3,54]. Furthermore, this SLR study also shows that assessments conducted by teachers build a positive impact in forming a solid coherence between theory and practice among students [55,61].

## 4. Discussion and Conclusions

This systematic review analyses the assessment approaches and the learning outcomes that are used to assess students in technical and vocational education. Overall, the results of the study have provided valuable new insights regarding students' learning outcomes in TVET assessment approaches. In particular, this study reveals that the findings of this study summarise the diversity of assessment approaches that have been implemented in TVET. The studies conducted differ depending on the objectives and results obtained by objective, method, and findings. Competency-based assessment is seen as the most widely used approach in TVET, showing 44%. The second most widely implemented assessment approach is performance-based assessment, at 18%, followed by formative assessment, at 11%, and other assessments at 3%. This is because assessment in TVET is more focused on competency building and student performance achievement [63]. This can be seen from the objective of the assessment that is to be implemented among them to see the next build the skills and competencies of students. Although the assessment approaches used are diverse, the objectives are closely related to assessing and developing students' skills, competencies, and competencies in various aspects [64].

Various teaching and learning methods are seen in TVET assessment studies, including problem-based learning [61], project-based learning [39], work-based learning [38] and mastery learning [50]. Assessment is also seen to be effectively implemented to assess student performance using technologies such as computers, such as the study by [44,45,56] versus conventional assessment. This study also shows that teachers play an essential role in ensuring the successful implementation of assessment. Teachers' skills and knowledge are critical indicators in student assessment. A study conducted by [59] showed that classroom-based assessments related to integrated learning work performed by teachers were ineffective and students were unable to relate theory and practise in the workplace. The authors of [61] study showed that although problem-based learning in assessment was successfully conducted, there were problems inherent in terms of the recruitment of unqualified TVET teachers.

The study of [36] also explained that the assessment system has not yet been able to assess the quality of students' competencies. TVET assessment in terms of building students' skills and competencies can assess various types of students' skills and competencies according to cognitive, psychomotor, and affective domains. Among them are employability skills, technical skills, generic skills, teamwork skills, problem-solving skills, leadership skills, and learning skills. Furthermore, TVET assessment is also seen to be able to have a positive impact on students in terms of building a positive attitude, training students for the real world of work, building community networks, and building potential in students [65]. In conclusion, the results show that TVET assessment tends to take the form of a competency assessment, with students' performance being the most desirable component. As it helps to measure students' characteristics, competency assessment is suited for use as a TVET assessment. Analysis from the SLR shows that assessment is viewed positively as support to students' learning and a mechanism for bridging the gap between current achievement and expected goals. It also shows that competency-based assessment is a type of assessment that is commonly used in technical and vocational fields.

## 5. Implication and Recommendations

This approach is critical to TVET since it contributes to the objective by producing competent and trained human resources. This systematic study provides information to students and stakeholders to build a positive environment that enhances student skills and competencies in learning. Actions should be taken to offer an assessment suitable that prioritises knowledge mastery, intellectual capital development, cultivating a progressive attitude culture, and supporting the practise of high virtue, ethics, and moral values. This study demonstrates the significance of conducting assessments using the proper assessment concepts and methods. Educators who conduct assessments should have a broad and in-depth understanding of the assessment 's methods, criteria, and expected outcomes. The accuracy of the information obtained from the assessment allows teachers, students, parents, and institutions to take appropriate action. Assessment in education not only examines students' abilities on the elements to be accomplished, but it also evaluates instructors' teaching approaches.

To achieve students' learning outcomes, teaching approaches and educators' assessment practises must also be prioritised. As a result, educators must prioritise their understanding of the suitable assessment techniques to be utilised, and they must master the skills in the implementation of such assessments in terms of proper processes, instruments, and work stages. Technology can also be used to improve the effectiveness of assessments. Classroom assessment, as proposed for further research, can be expanded in its implementation. The gap in these systematic reviews is identified as classroom assessment. This is due to the fact that assessment in the classroom is viewed as a holistic evaluation that incorporates summative and formative tests appropriate for measuring and developing students' knowledge, abilities, and positive values. The next follow-up study will look at how assessment can be used to build students' skills in the industry 4.0 revolution, as no studies have been conducted to assess and build these skills based on this systematic literature review. This is significant because, in order to lead the industry's 4.0 revolution, students must be equipped with the skills necessary to compete in the twenty-first century.

**Author Contributions:** S.R.M.Y., M.S.R. and R.M.Y. came up with the idea for this article; S.R.M.Y., M.S.R., R.M.Y. and H.U.H. performed the literature search and data analysis; S.R.M.Y., M.S.R., R.M.Y. and H.U.H. drafted and/or critically revised the work; S.R.M.Y., M.S.R., R.M.Y. and H.U.H. were responsible for writing—review and editing; S.R.M.Y., M.S.R., R.M.Y., N.A.J. and H.U.H. read and approved the final manuscript. All authors have read and agreed to the published version of the manuscript.

**Funding:** This research and the APC was funded by Enculturation Research Centre, Faculty of Education, Universiti Kebangsaan Malaysia, grant number PDE52.

**Institutional Review Board Statement:** Not Applicable.

**Informed Consent Statement:** Not Applicable.

**Acknowledgments:** Not applicable.

**Conflicts of Interest:** The authors declare no conflict of interest.

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
