# Peer review of "An Assessment Approaches and Learning Outcomes in Technical and Vocational Education: A Systematic Review Using PRISMA"

_sustainability, doi:10.3390/su14095225_

Round 1

Reviewer 1 Report

The paper have a good subject and it is very good explained by the result pf the research.  Succes!

Author Response

Thank you for your kind review. I will be looking forward to my paper being published soon.

Reviewer 2 Report

Dear all

I have added my review as a PDF attachment.

Anonymous Reviewer

Author Response

Dear reviewer

We improved the SLR article in response to your feedback. It has been changed in accordance with the PRISMA criteria for article preparation. Because the improvements cover the majority of the parts, you may look at the article as a whole.

Thank you very much.

Reviewer 3 Report

The idea is very current and useful! We propose some suggestions for improving the study:

  1. There is no research questions, purpose and/or hypothesis and objectives- we suggest their elaboration (maybe accordingly with results from Table 3,4,5,6)  and the correlation of the analysis results accordingly.
  2. The research design must be improved: description of type of research (quantitative, qualitative, mixed) an how it was done-an additional description; how it was used in the research MMAT, how and why certain studies were selected for QN, or QL or MM; specifying the period in which the research was performed; 
  3. The statistical component of the research is not enough relevant;
  4. The final number of studies analyzed is relatively small;
  5. It may be useful to report the results to the conditions in the education system in which the research was conducted;

Author Response

Thank you for your kind review. Please refer to the attachment file. 

Round 2

Reviewer 2 Report

Review for the manuscript ”Analysis of the Assessment Approaches and Outcomes Learning in Technical and Vocational Education”

Journal: Sustainability

Round: 2 (1)

Date: 21.3.2022

This is the second review round for the paper. Now I see that authors have revised the MS according to PRISMA guidelines.

This can be published after minor revisions. Here are my additional questions / comments:

  1. The RQ 1 clear but the RQ2 is not:
    1. What are the intended learning outcomes? in the assessment conducted?
    2. Is the second sentence full?
  2. In the line 83 you write “Furthermore, there is a requirement to provide current knowledge in order to ensure that significant insights can be derived from the existing literature.”
    1. Please justify more precisely via examples and citations. This is important why?
  3. The same for lines 88-98 where you justify the syst. review. Please cite and argument via methodological papers.
  4. Why did you select this timeline?
  5. In results section, indicate which sections answer to RQ1 and which to RQ2.
  6. Add an extra column to table 1 that indicates the number of the paper. It should be matching to 29.

Author Response

Response to Reviewer 2 for 2nd revision

Dear Reviewer,

Refer to your comment: This is the second review round for the paper. Now I see that authors have revised the MS according to PRISMA guidelines.This can be published after minor revisions. Here are my additional questions / comments:

  1. The RQ 1 clear but the RQ2 is not: What are the intended learning outcomes? in the assessment conducted? Is the second sentence full?

A: I provided the new version of RQ 2 for this study as indicated at line 72

  1. In the line 83 you write “Furthermore, there is a requirement to provide current knowledge in order to ensure that significant insights can be derived from the existing literature.” Please justify more precisely via examples and citations. This is important why?

A: I provided citation that is indicates the importance of this SLR study performed in TVET. It was indicated in line 85 to 94.

  1. The same for lines 88-98 where you justify the syst. review. Please cite and argument via methodological papers.

A: I provided a citation that indicates the importance of this SLR study performed according to methodological papers. It was indicated in line 96 to 105.

  1. Why did you select this timeline?

A: I was indicated the time span of selected publications published as in lines 186 to 190.

  1. In results section, indicate which sections answer to RQ1 and which to RQ2.

A: I provided RQ 1 in lines 336 to 337 and RQ 2 in lines 539 to 541 for the result section.

  1. Add an extra column to table 1 that indicates the number of the paper. It should be matching to 29.

A: I added an extra column to table 1 that indicates the number of the paper as your suggestions

Thank you for your kind attention. 

Reviewer 3 Report

Thank you for submitting the new version of the paper. It look much more consistent and complete. It will be useful only a more precise reformulating of the research objectives. Good luck!

Author Response

Response to Reviewer 3 for 2nd revision

Reviewer 3 comment: Thank you for submitting the new version of the paper. It look much more consistent and complete. It will be useful only a more precise reformulating of the research objectives. Good luck!

Response: Thank you for your comment. I provided the objective of this study that indicated in line 62 to 65.

The purpose of this review paper is to provide an overview of TVET assessment. The researchers intended to delve deeper into the assessment approaches used by educators and the learning outcomes expected from the assessment, particularly in terms of developing student competencies.  We are also interested in the assessment technique that are often implemented TVET which is significant with the current challenges in research on education.

I hope this response meets your requirement. 

Thank you very much

This manuscript is a resubmission of an earlier submission. The following is a list of the peer review reports and author responses from that submission.